# Association between Extended Meropenem Regimen and Achievement of Aggressive PK/PD in Patients Receiving Continuous Renal Replacement Therapy for Septic AKI

**DOI:** 10.3390/antibiotics13080755

**Published:** 2024-08-11

**Authors:** Shinya Chihara, Tomoyuki Ishigo, Satoshi Kazuma, Kana Matsumoto, Kunihiko Morita, Yoshiki Masuda

**Affiliations:** 1Department of Intensive Care Medicine, Sapporo Medical University, School of Medicine, Sapporo 060-8543, Japan; chr.19720324@me.com (S.C.); miminekko@me.com (Y.M.); 2Department of Clinical Engineering, Japan Health Care University Faculty of Health Sciences, Sapporo 062-0053, Japan; 3Department of Pharmacy, Sapporo Medical University Hospital, Sapporo 060-8543, Japan; ishigo@sapmed.ac.jp (T.I.); 4Department of Clinical Pharmaceutics, Faculty of Pharmaceutical Sciences, Doshisha Women’s College of Liberal Arts, Kyotanabe 610-0395, Japan; kmatsumo@dwc.doshisha.ac.jp (K.M.); kmorita@dwc.doshisha.ac.jp (K.M.)

**Keywords:** meropenem, critically ill patients, continuous renal replacement therapy, acute kidney injury, pharmacokinetics/pharmacodynamics analysis, therapeutic drug monitoring

## Abstract

Aggressive pharmacokinetic (PK)/pharmacodynamic (PD) targets have shown better microbiological eradication rates and a lower propensity to develop resistant strains than conservative targets. We investigated whether meropenem blood levels, including aggressive PK/PD, were acceptable in terms of efficacy and safety using a meropenem regimen of 1 g infusion every 8 h over 3 h in patients undergoing continuous renal replacement therapy (CRRT) for septic acute kidney injury (AKI). Aggressive PK/PD targets were defined as the percentage of time that the free concentration (%fT) > 4 × minimal inhibitory concentration (MIC), the toxicity threshold was defined as a trough concentration >45 mg/L, and the percentage of achievement at each MIC was evaluated. The 100% fT > 4 × MIC for a pathogen with an MIC of 0.5 mg/L was 89%, and that for a pathogen with an MIC of 2 mg/L was 56%. The mean steady-state trough concentration of meropenem was 11.9 ± 9.0 mg/L and the maximum steady-state trough concentration was 29.2 mg/L. Simulations using Bayesian estimation showed the probability of achieving 100% fT > 4 × MIC for up to an MIC of 2 mg/L for the administered administration via continuous infusion at 3 g/24 h. We found that an aggressive PK/PD could be achieved up to an MIC of 0.5 mg/L with a meropenem regimen of 1 g infused every 8 h over 3 h for patients receiving CRRT for septic AKI. In addition, the risk of reaching the toxicity range with this regimen is low. In addition, if the MIC was 1–2 mg/L, the simulation results indicated that aggressive PK/PD can be achieved by continuous infusion at 3 g/24 h without increasing the daily dose.

## 1. Introduction

The Japanese Guidelines for the Treatment of Sepsis and Septic Shock 2020 (J-SSCG 2020) recommend early antimicrobial treatment once sepsis or septic shock is confirmed [1]. Meropenem (MEPM), a time-dependent carbapenem, is a broad-spectrum antimicrobial with potent antimicrobial activity against Pseudomonas aeruginosa and anaerobic bacteria and is recommended for empiric therapy in severe sepsis [2]. For beta-lactam antibiotics, including meropenem, conservative pharmacokinetic (PK)/pharmacodynamic (PD) targets 40–50% or 100% of the time that the free concentration (%fT) > minimal inhibitory concentration (MIC) has traditionally been considered sufficient for clinical efficacy [3]. However, aggressive PK/PD targets with Cmin/MIC ratios of 4–5 or higher as cutoff levels have better micro-biological eradication rates and a lower tendency to develop resistant strains than conservative targets [4,5,6]. An MEPM dosing design that targets aggressive PK/PD may be helpful in preventing microbiological failure and resistance in critically ill patients. On the other hand, although MEPM is an antimicrobial agent with a relatively broad safety zone, it can still have undesirable effects; plasma trough concentrations of MEPM exceeding 64.2 mg/L and 44.5 mg/L, respectively, may result in neurotoxicity and nephrotoxicity [7].

Sepsis is often associated with acute kidney injury (AKI) and requires the initiation of continuous renal replacement therapy (CRRT) in approximately 70% of patients [8]. Expected therapeutic benefits of CRRT include the removal of various sepsis-related mediators to increase the likelihood of multidisciplinary management and renal support, and the removal of electrolytes and uremia-related substances [9,10]. However, the dialysate flow rate (QD) and filtration flow rate (QF) of CRRT reportedly increase, in turn removing useful substances such as drugs, nutrients, electrolytes, and trace elements [11]. Although appropriate antibiotic administration is crucial for achieving optimal bactericidal efficacy and improving clinical outcomes in critically ill patients undergoing CRRT [12], the PK in critically ill patients undergoing CRRT is complex. When CRRT is required, the total MEPM clearance becomes greatly reduced as renal function declines, but the rate of extracorporeal removal is higher owing to physicochemical properties, such as low molecular weight and low plasma protein binding [13]. Furthermore, the type and intensity of CRRT in terms of flow rate can affect MEPM clearance [14]. Additionally, the patient’s residual renal function may be appropriate [15,16,17]. Many studies have examined the PK/PD characteristics of MEPM in CRRT patients [15,16,17,18,19,20,21], but each model has different covariates and different recommended doses. The model by Burger et al. [18] included total CRRT flow as a structural covariate of clearance and 100% fT > MIC was used as a measure of efficacy, while other models did not include CRRT parameters such as flow or dialysis type as covariates [15,16,17,19,20,21]. Further covariates on clearance were described by either residual diuresis [15,16] or estimated glomerular filtration rate (*eGFR*) [17]. In addition, the predictive performance of pre-administration between models differs substantially [22]. Aggressive PK/PD goals for β-lactams, including MEPM, have been achieved at a high rate with prolonged or continuous infusion [23,24]. However, most previous studies have created population models based on blood levels of MEPMs administered for short durations, and the actual rate of achieving target PK/PD with prolonged infusion has not been determined. In addition, dosing regimens targeting aggressive PK/PD in patients receiving CRRT for septic AKI have not been fully explored.

Therefore, we investigated whether MEPM blood levels, including aggressive PK/PD, are acceptable in terms of efficacy and safety using a MEPM regimen of 1 g infusion every 8 h over 3 h in patients undergoing CRRT for septic AKI. In addition, we evaluated the achievement of aggressive PK/PD in different MEPM dosing regimens and different MICs using Bayesian estimation and Monte Carlo simulation.

## 2. Material and Methods

### 2.1. Patients

This study was approved by the Ethics Review Committee of Sapporo Medical University (approval no. 23–37). The subjects were 9 patients who underwent CRRT for septic AKI in our intensive care unit between October 2011 and May 2013. Patients younger than 15 years of age, those who had received blood purification therapy other than CRRT, those with hematologic malignancies, and those receiving immunosuppressive therapy were excluded. Informed consent was obtained from all patients and their families, and CRRT was administered for at least 24 h after consent was obtained. Patients were treated with MEPM 1 g every 8 h over 3 h after the start of CRRT, and the criteria for the induction of continuous venovenous filtration (CVVH) were stage 1 or higher according to the Kidney Disease Improving Global Outcomes (KDIGO) criteria for AKI [25]. Septic shock was diagnosed using the previous diagnostic criteria that met the current sepsis-3 criteria [26]. Standard treatment consisted of total culture (including blood cultures), lactate monitoring, early massive infusion within 6 h, aggressive treatment of infected foci, drainage, and antibiotics after other procedures, according to the Surviving Sepsis Campaign 2012 guidelines [27].

### 2.2. Collection of Patient Data

Patient background data including age, sex, Acute Physiology and Chronic Health Evaluation II (APACHE II), and Sequential Organ Failure Assessment (SOFA) scores were collected. Serum creatinine (SCr) level, *eGFR*, and creatinine clearance (CLcr) were assessed before CRRT initiation.

### 2.3. Preparation of Extracorporeal Circulation by CRRT

The blood flow rate (QB) of the CRRT was set at 150 mL/min, and the filtration flow rate (QF) was set at 1000 and 2000 mL/h. CRRT was performed with TR-55X device (Toray Medical, Tokyo, Japan), hemofilter: 1.3 m^2^ polysulfone (Excelflo^®^AEF-13, Asahi Kasei Medical, Tokyo, Japan), hemofilter: JCH-SMU (Japan Lifeline, Tokyo, Japan), blood circuit: SUBPACK^®^ (Nipro, Osaka, Japan). The anticoagulant used for CRRT was nafamostat mesylate, administered at 20–30 mg/h.

### 2.4. Determination of Plasma MEPM Concentrations during CRRT

A 1 g dose of MEPM was administered intravenously as a 3 h infusion every 8 h. MEPM concentrations were measured after the start of administration, and concentrations of MEPM were assessed at 3, 5, 8, 11, 13, 16, 19, 21, and 24 h after the start of MEPM administration. Blood samples (2 mL) were collected in test tubes containing 0.9% citrate, and sampling was performed at two sites before and after hemofiltration. Blood samples were centrifuged, and only plasma samples were collected in tubes containing 3-(N-morpholino)-propanesulfonic acid (MOPS) buffer (pH 7.0) and stored at −80 °C until assayed. The MOPS buffer was added as a MEPM stabilizer in plasma. The activity of MEPM in plasma mixed with the same volume of 1 M MOPS buffer (pH 7.0) was stable at −20 °C for 3 days and −80 °C for 2 months [28]. The samples were processed by injecting all 300 µL of plasma into an ultrafiltration device (Centrifree YM-30, Millipore). Plasma MEPM concentration was determined by injecting 30 µL of the filtrates obtained by centrifugation of the ultrafiltered plasma samples at 3000 rpm for 5 min into a high-performance liquid chromatography (HPLC) system. The HPLC instrument had an LC-20AB pumping unit and SPD-20A (Shimadzu Corporation, Kyoto, Japan) ultraviolet-visible spectrophotometric detector. The mobile phase was a mixture of PIC-A reagent/methanol (75:25), the flow rate was 1.0 mL/min, the detection wavelength was 300 nm, and the column was HYPERSIL ODS-5 (4.6 mm, I.D. × 250 mm; Chemco Scientific Co., Ltd., Tokyo, Japan). For the assay using this system, the retention time of MEPM was 12.5 min, the lower limit of detection was 0.1 µg/mL, and good linearity was obtained in the range of 0.1 to 200 mg/L, with intraday and interday variations being within 5%. The CVVH clearance (CL_CVVH_) of MEPM was measured using a sample of three peaks at the time of MEPM administration during CRRT. The calculated clearance was compared between the QF of 1000 mL/h and 2000 mL/h. The clearance of MEPM was calculated as previously reported, using the following equation [29]:CL_CVVH_ (L/h) = (Cti − Cto)/Cti × (QT − QF) + QF (Cti and Cto: concentration of substances before and after hemofilter, respectively).

### 2.5. PK/PD Parameter Calculation

Several PopPK analyses of MEPM in CRRT patients have been reported, and different factors have been incorporated as covariates and other factors [22]. Recent reports have investigated the accuracy of PopPK parameters of the MEPM in patients undergoing CRRT [22]. In the present study, the number of patients with residual renal function was large, and Bayesian estimation was performed using multiple blood concentration points based on the population model of Niibe et al. (Equation (1)) [17].
(1)$$Total\;clearance\;({CL}_{total})=4.22\times {\left(eGFR/26.1\right)}^{0.25}$$

Other PK parameters were as follows: inter-compartmental clearance, 7.84 L/h; central volume of distribution (V1), 14.82 L; peripheral volume of distribution (V2), 11.75 L; ωCL, 26.75%; ωV1, 27.33%; ωV2, 57.05%; proportional error, 18.39%. Pharmacokinetic parameters in this study were evaluated using Bayesian estimation. To assess the validity of the population model used in this study, R^2^, absolute error (mean squared error; RMSE), and relative error (mean absolute percentage error; MAPE) were evaluated for the predictions from Bayesian estimation and the measured meropenem plasma concentrations. In addition, we also evaluated the percentage of differences between the measured and predicted concentrations obtained by Bayesian estimation that were <2 standard deviations (SDs).

According to the Clinical Laboratory Standards Institute (CLSI), the MIC breakpoint for *Enterobacter* spp. is 1.0 mg/L, and the susceptible MIC breakpoints for *Pseudomonas aeruginosa* and *Acinetobacter baumannii* are 2.0 mg/L, for the pathogen MIC of 2 mg/L, special emphasis was placed on achieving the targets of 40% fT > MIC, 100% fT > MIC, 40% fT > 4 × MIC, and 100% fT > 4 × MIC [30]. In addition, the rate of % fT > MIC achievement at other MICs (0.25, 0.5, 1, 2, 4, 8, 16, 32, and 64 mg/L) for each patient was investigated. The toxicity threshold was defined as a steady-state trough concentration > 45 mg/L [7].

### 2.6. Dosing Simulations

Monte Carlo simulations were performed after inputting the patients’ covariates using BMs-Pod (version 8.06). For each scenario, 5000 virtual subjects were simulated. After entering the patient covariates (age, sex, body weight, and SCr), the model predicted the proportion of aggressive PK/PD achieved under different PK parameters and MICs. Based on the patient background of the study, the mean age, body weight, and SCr were set at 63 years, 70.2 kg, and 3.4 mg/dL, respectively. The male-to-female ratio was set to be 1:1. Aggressive PK/PD targets with 100% fT > 4 × the MIC were used as surrogate markers [4,5,6]. The empirical MEPM dosing regimens of 1 g infused every 12 h over 30 min, 1 g infused every 8 h over 3 h, and 3 g infused over 24 h were individually simulated. Based on the predicted meropenem concentrations at a steady state, a PK/PD probability of target attainment (PTA) of 90% or greater was considered desirable.

### 2.7. Statistical Analysis

Continuous variables in patient characteristics were represented using mean ± standard deviation (SD) (min–max) and analyzed using the Mann–Whitney U-test. Pearson’s correlation coefficient was used to examine correlations for each continuous variable. *p* < 0.05 was considered statistically significant.

## 3. Results

### 3.1. Patient Characteristics

Patient characteristics are shown in Table 1. The mean age of the patients was 63.4 ± 12.7 years. Of the patients, 22% were female, and the mean body weight was 70.2 ± 14.1 kg. The mean APACHE II score of the patients was 25.4 ± 10.9, and the SOFA score was 9.6 ± 3.1. Six of the nine patients had septic shock and were under ventilatory management. Urine output was 1085 ± 800 mL/day, SCr was 3.4 ± 3.2 mg/dL, and *eGFR* was 26.3 ± 10.3 mL/min.

### 3.2. Pharmacokinetic Parameters of MEPM

The estimated pharmacokinetic parameters of MEPM are presented in Table 2. The mean steady-state trough concentration of MEPM was 11.9 ± 9.0 mg/L. The maximum concentration of MEPM was 29.2 mg/L, and none of the samples exceeded 45 mg/L (Table 2). The mean *CL_total_* of MEPM was 6.6 ± 3.3 (3.0–13.9) L/h and V_total_ was 26.6 ± 5.9 (19.6–38.0) L. Although there was a positive tendency between *eGFR* and *CL_total_*, it was not statistically significant (Figure 1a). Similarly, there was a negative tendency between *eGFR* and CL of CVVH (Figure 1b). There was no significant difference in the *CL_total_* of MEPM between the QF 1000 mL/h and QF 2000 mL/h groups (5.6 ± 0.8 L/h vs. 7.4 ± 4.4 L/h, *p* = 0.713). Similarly, there was no significant difference in CL_CVVH_ between the two groups (0.8 ± 0.9 L/h vs. 0.5 ± 0.5 L/h, *p* = 0.391). There were no significant differences between the two groups in V1, V2, and total systemic distribution volume (V_total_). The predictions obtained by Bayesian estimation versus observations of all measured meropenem plasma concentration plots for the population model used in this study were unbiased, with an accuracy and precision of RMSE 1.771 and MAPE 10.5%, respectively (Figure 2a). In addition, the difference between predicted and measured concentrations was roughly in the range of ± 2 SD at all blood sampling points (96%: 78/81), regardless of the measured concentration (Figure 2b).

### 3.3. Percentage of Target MIC Achieved and %TAM 

The results of 40% fT > MIC, 100% fT > MIC, 40% fT > 4 × MIC, and 100% fT > 4 × MIC for various MEPM MICs are shown in Figure 3. The two typical targets for pathogen MICs at 2 mg/L (40% and 100% fT > MIC) were 100% (9/9) and 89% (8/9), respectively (Figure 3). The % time above MIC (%TAM) decreased with increasing MIC, from 97.9 ± 6.2% at MIC 2 mg/L to 65.2 ± 28.7% at MIC 16 mg/L (Table 3).

For the percentage of aggressive targets achieved, the probabilities of achieving 40% fT > 4 × MIC and 100% fT > 4 × MIC for pathogen a MIC of 2 mg/L was 100% (9/9) and 56% (5/9), respectively (Figure 3). Simulations using Bayesian estimation showed the probability of achieving 100% fT > 4 × MIC up to an MIC 2 mg/L for the administered via continuous infusion at 3 g/24 h (Figure 4).

### 3.4. Simulation and the Probability of Achieving the Target in Each MIC and the Recommended Dosing Regimen

For a PK/PD target of 100% fT > 4 × MIC (Figure 5), almost all simulated regimens reached desirable PTA levels (PTA levels of >90%) against pathogens with an MIC of 1 mg/L. Only 1 g every 12 h failed to obtain satisfactory PTAs in patients infected with pathogens with MICs of 2 mg/L. The optimal PTA for pathogens with an MIC greater than 4 mg/L could only be achieved by infusing 3 g over a 24 h period.

## 4. Discussions

The results of this study indicate that a meropenem regimen of 1 g infusion every 8 h over 3 h in patients undergoing CRRT for septic AKI has a high probability of achieving an aggressive PK/PD target of 40% fT > 4 × MIC for bacterial species up to MIC 2 mg/L and 100% fT > 4 × MIC up to MIC 0.5 mg/L. On the other hand, achieving an aggressive PK/PD target of 100% fT > 4 × MIC for MIC ≥1 mg/L may be difficult with this dosing regimen, in which case, continuous infusion [31] may be a useful option (Figure 4 and Figure 5). Furthermore, 1 g every 8 h over the 3 h regimen in this study was less likely to result in trough concentrations above 45 mg/dL, a risk factor for toxicity [7], suggesting that this regimen is safe in CRRT patients with residual renal function. In fact, there were no cases in which the clinical diagnosis revealed adverse effects that could be attributed to MEPM, although EEG was not performed. Additionally, the slight difference observed in the results of the Monte Carlo simulation suggests that in patients with residual renal function, blood levels should be monitored to assess dose adequacy.

The strength of this study is the Bayesian estimation of PK/PD from nine MEPM blood concentrations, including the trough and peak, in various CRRT patients in real-world clinical settings. It has been reported that the accuracy of prediction increases as the number of blood concentration sampling points increases [30], which provides an accurate PK/PD ratio and clarifies the range of effective MICs and the safety of extended MEPM administration in CRRT patients with residual renal function.

Previous reports have shown that in non-traumatic CRRT patients with anuria, a regimen of 1 g of MEPM every 8 h over 3 h can achieve 100% fT > MIC if the MIC is 4 mg/L or less [31], and a regimen of 2 g every 8 h for 3 h is associated with a risk of toxicity [31]. As residual renal function was observed in this study, it is possible that there was a slight difference in the probability of achieving an MIC of 4 mg/L compared to previous reports. In contrast, the MIC to which *Pseudomonas aeruginosa* and other bacteria are considered susceptible according to the CLSI criteria is 2 mg/dL, and a regimen of 1 g every 8 h for 3 h is considered sufficient to achieve a 100% fT > MIC (Table 3 and Figure 3). In addition, if the MIC was 1–2 mg/L, the simulation results indicated that 100% fT > 4 × MIC can be achieved by continuous infusion at 3 g/24 h without increasing the daily dose (Figure 4).

Another possible strategy to achieve the target MIC is to increase the MEPM dose. However, previous reports have indicated that MEPM dosing regimens of 2 g every 6–8 h for short-term infusion, 2 g every 8 h for long-term infusion, and 6 g every 24 h for continuous infusion are not recommended for critically ill CRRT patients because of the expected high toxicity risk [31]. The CL and Vd levels of the MEPM obtained in this study varied considerably among the individuals (Table 2). Previously reported CLs for non-traumatic CRRT patients or in critically ill patients were approximately 3 L/h, but they were widely reported to be approximately 9 L/h in severe non-CRRT patients and 15.5 L/h in healthy subjects [23,31,32]. Although this study included only CRRT patients, they were distributed over this wide range. The volume of distribution was approximately 28 L, which is more than 21 L in healthy subjects and approximates the levels in previous CRRT patients [31]. In critically ill patients undergoing CRRT, the increase in the volume of distribution can be explained by the fluid overload associated with sepsis-induced capillary leak syndrome [32,33].

It has been reported that CRRT removes both the etiological agent and the administered drug out of therapeutic necessity and that the target therapeutic concentration cannot be maintained [34,35]. MEPM has a molecular weight of approximately 400 Da, volume distribution of 0.37 L/kg, and protein binding rate of <2%, which is very low; therefore, it can be easily removed using CRRT. Although the maintenance of blood levels of MEPMs with broad-spectrum antimicrobial agents is an important component of the treatment strategy for sepsis, there are mixed reports regarding the removal of MEPMs by CRRT, with some reporting that removal is possible and others reporting that CRRT has little impact [15,16,17,18,19,20,21,36,37]. In this study, the clearance of CVVH was small compared to the *total clearance*, and the degree of influence was considered limited if the QF was approximately 1000–2000 mL/h (Table 2 and Figure 1). On the other hand, when CRRT is performed for septic AKI, CRRT with increased dialysate flow/filtration flow is performed for the following reasons: to increase the efficiency of removal of large molecular weight substances such as cytokines and to correct metabolic acidosis caused by increased tissue oxygen demand. Therefore, when performing CRRT at a higher flow rate for septic AKI with shock, it may be necessary to consider the removal of antimicrobial agents, measurement of blood levels, and design the administration regimen accordingly.

### Limitations

Our study has several limitations. First, the small sample size may have led to underestimation or overestimation of the data. Second, the KDIGO guidelines [25] recommend 20–25 mL/kg/h as the optimal dialysis volume for CRRT for AKI. However, the current status of CRRT for AKI in Japan differs from that of overseas countries, with 10–15 mL/kg/h as the usual dose. This dose is only approximately half of the guideline recommendation, which is largely due to the fact that this is the maximum amount of dialysis allowed by insurance. Therefore, considering the KDIGO guidelines and the situation in Japan, our institution uses a maintenance flow rate of approximately 1000–2000 mL/h for CRRT (20–40 mL/kg/h for a 50 kg body weight), which is not based on body weight conversion. The effect of this difference in flushing flow rates between Japan and other countries is not clear, and large-scale studies are needed to address these issues. Therefore, large-scale studies addressing these issues are warranted.

## 5. Conclusions

We found that an aggressive PK/PD could be achieved up to an MIC of 0.5 mg/L with a MEPM regimen of 1 g infused every 8 h over 3 h for patients receiving CRRT for septic AKI. In addition, the risk of reaching the toxicity range with this regimen is low. In addition, if the MIC was 1–2 mg/L, the simulation results indicated that 100% fT > 4 × MIC can be achieved by continuous infusion at 3 g/24 h without increasing the daily dose. Although there was a positive tendency between *eGFR* and CL of MEPM, owing to the complexity of the patient’s background, it is desirable to monitor blood concentrations in each case and design the meropenem regimen accordingly. In addition, this study requires future validation using additional cases.

## Figures and Tables

**Figure 1 antibiotics-13-00755-f001:**
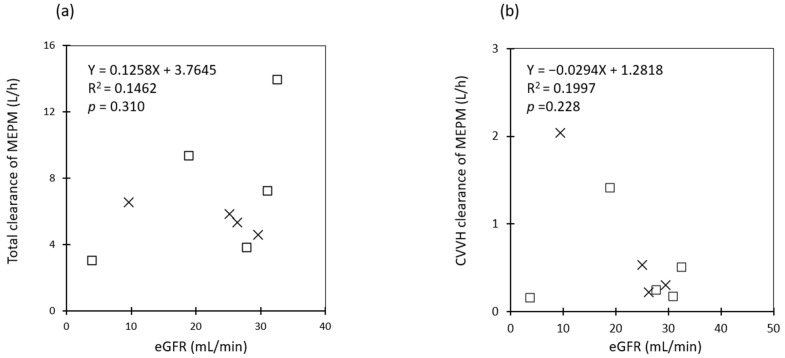
Correlation between *eGFR* and meropenem clearance. □: QF of 2000 mL/h (n = 5), ×: QF of 1000 mL/h (n = 4). (**a**) Plot correlation between *eGFR* (mL/min) and *total clearance* of MEPM (L/h), (**b**) Plot correlation between *eGFR* (mL/min) and MEPM clearance of CVVH (L/h). Abbreviations: MEPM, meropenem; QF, quantity of filtration flow rate; *eGFR*, estimated glomerular filtration rate; CVVH, continuous venovenous filtration.

**Figure 2 antibiotics-13-00755-f002:**
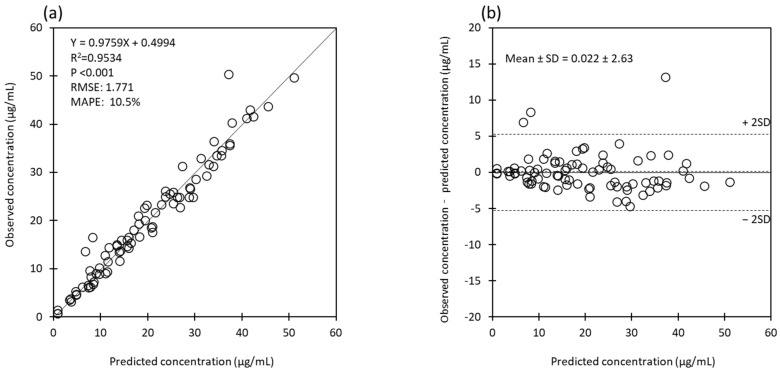
(**a**) Predictions obtained by Bayesian estimation versus observations of all measured meropenem plasma concentrations. Black line: line of identity. (**b**) The difference between the predictions obtained by Bayesian estimation and the observations of all measured meropenem plasma concentrations. Dotted lines indicate mean and ± 2 SD. Abbreviations: RMSE, root mean squared error; MAPE, mean absolute percentage error; SD, standard deviation.

**Figure 3 antibiotics-13-00755-f003:**
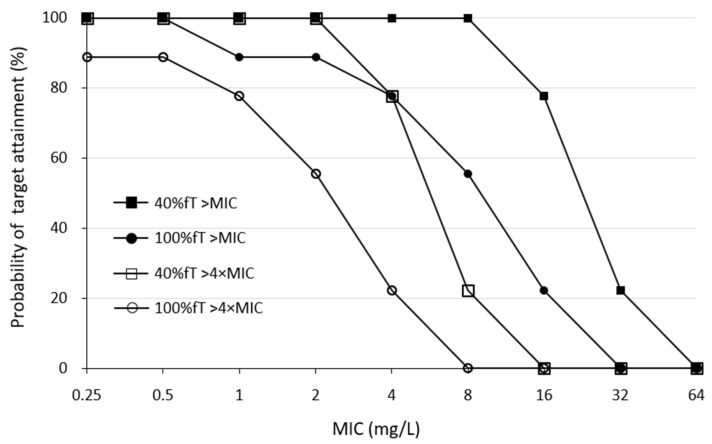
Percentage of patients achieving target PK/PD in each MIC. Abbreviations: MEPM, meropenem; MIC, minimal inhibitory concentration; PK, pharmacokinetic; PD, pharmacodynamic.

**Figure 4 antibiotics-13-00755-f004:**
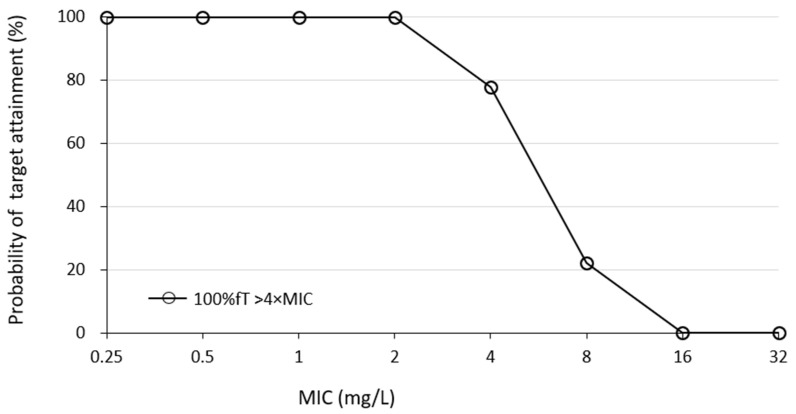
Percentage of cases achieving target PK/PD at each MIC when simulations using Bayesian estimation with 3 g infused over 24 h. Abbreviations: MEPM, meropenem; MIC, minimal inhibitory concentration; PK, pharmacokinetic; PD, pharmacodynamic.

**Figure 5 antibiotics-13-00755-f005:**
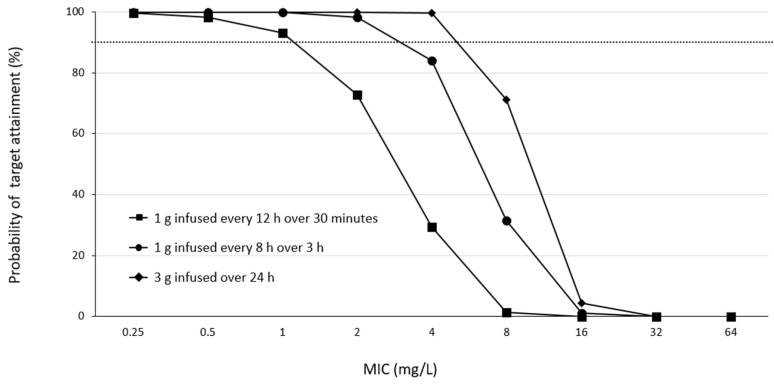
PTA versus MIC profiles for various simulated MEPM dosing regimens based on the PK/PD targets of 100% fT > 4 × MIC. Dashed horizontal lines represent a PTA of 90%. Abbreviations: MEPM, meropenem; PTA, probability of target attainment; MIC, minimal inhibitory concentration; PK, pharmacokinetics; PD, pharmacodynamics.

**Table 1 antibiotics-13-00755-t001:** Patient characteristics.

	n = 9
Age, years	63 ± 13 (46–80)
Female, n (%)	2 (22%)
Height, cm	162.7 ± 8.7 (147.0–175.3)
Body weight, kg	70.2 ± 14.1 (47.7–95.0)
BMI, kg/m^2^	26.4 ± 3.9 (19.5–30.9)
SOFA score	9.5 ± 3.0 (5–13)
APACHE II score	25.4 ± 10.8 (12–50)
Laboratory data	
Albumin, g/dL	2.5 ± 0.4 (2.2–3.3)
Creatinine, mg/dL	3.4 ± 3.2 (1.2–10.6)
CLcr, mL/min	41.1 ± 19.8 (7.7–70.7)
*eGFR*, mL/min	22.8 ± 10.0 (4.0–32.8)
Dialysis type	
CVVH, n (%)	9 (100%)
Total flow rate (mL/h)	1556 ± 527 (1000–2000)
Residual diuresis, mL/24 h	1085 ± 800 (106–2646)

Data are presented as mean ± SD (min–max) or number (with percentage). BMI, body mass index; SOFA, Sequential Organ Failure Assessment; APACHE II, Acute Physiology and Chronic Health Evaluation II; CLcr, creatinine clearance; *eGFR*, estimated glomerular filtration rate; CVVH, continuous venovenous filtration.

**Table 2 antibiotics-13-00755-t002:** Pharmacokinetic parameters estimate for meropenem.

	All (n = 9)	QF of 1000 mL/h (n = 4)	QF of 2000 mL/h (n = 5)	*p*-Value
Steady-state trough concentration (mg/L)	11.9 ± 9.0 (0.9–29.2)	10.7 ± 3.0 (8.0–14.2)	12.8 ± 12.4 (0.9–29.2)	0.713
Predicted Css (mg/L) *	22.7 ± 9.9 (9.0–41.2)	22.6 ± 3.4 (18.9–27.0)	22.8 ± 13.7 (9.0–41.2)	0.624
*CL_total_* (L/h)	6.6 ± 3.3 (3.0–13.9)	5.6 ± 0.8 (4.6–6.6)	7.4 ± 4.4 (3.0–13.9)	0.713
CL_CVVH_ (L/h)	0.6 ± 0.7 (0.2–2.0)	0.8 ± 0.9 (0.2–2.0)	0.5 ± 0.5 (0.2–1.4)	0.391
V1 (L)	15.0 ± 2.0 (12.4–18.6)	13.9 ± 1.0 (12.4–14.8)	15.9 ± 2.3 (13.4–18.6)	0.391
V2 (L)	11.5 ± 4.2 (6.2–19.3)	10.5 ± 3.4 (7.8–15.3)	12.4 ± 4.9 (6.2–19.3)	0.713
V_total_ (L)	26.6 ± 5.9 (19.6–38.0)	24.4 ± 3.9 (20.2–29.4)	28.3 ± 7.1 (19.6–38.0)	0.540

Data are presented as mean ± SD (min–max). QF, filtration flow rate; *CL_total_*, *total clearance*; V1, central volume of distribution; V2, peripheral volume of distribution; V_total_, total volume of distribution. * Predicted steady-state blood concentration assuming a continuous infusion of 3 g/24 h.

**Table 3 antibiotics-13-00755-t003:** %TAM for each MIC.

MIC	0.25	0.5	1	2	4	8	16	32	64
%TAM	100	100	99.6 ± 1.2	97.9 ± 6.2	95.6 ± 12.0	89.7 ± 20.3	65.2 ± 28.7	21.8 ± 29.6	0

Abbreviations: %TAM, % time above MIC; MIC, minimum inhibitory concentration.

## Data Availability

The datasets generated and/or analyzed during this current study are not publicly available because a research agreement from all authors is required for data sharing; however, they are available from the corresponding author on reasonable request.

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
