# Peer review of "Association between Extended Meropenem Regimen and Achievement of Aggressive PK/PD in Patients Receiving Continuous Renal Replacement Therapy for Septic AKI"

_antibiotics, 2024, doi:10.3390/antibiotics13080755_

Round 1
Reviewer 1 Report
Comments and Suggestions for Authors
In the manuscript "Association between Extended Meropenem Regimen and Achievement of Aggressive PK/PD in Patients Receiving Continuous Renal Replacement Therapy for Septic AKI", the authros aimed to investigate the safety and efficacy of a specific meropenem regimen in patients undergoing CRRT for septic AKI. Although the manuscript withholds valuable data regarding safety and efficacy of meropenem in patients undergoing CRRT for septic AKI, it needs to undergo major revisions before considered for publication.
Major
1. Abstract/ Introduction should be revised to underline the significance (novelty) of this study and to include appropriate aim.
- In abstract (last couple sentences), introduction (last paragraph), and also conclusion, large part of the study is missing -- simulations. Authors should revised abstract, introduction, and conclusion so that they would encompass the whole study.
- More importantly, the significance (novelty) of this study should be sufficiently explained in introduction. In the current version of the manuscript, "Although appropriate .. aggressive targets, are being achieved", the summary of the previous studys and their limitations are too concise for readers to understand this study's importance.
Authors should extend on the summary of the previously published literatures on the topic and explain the novelty of this study.
2. Methods
- It seems that method section missing large piece of information especially regarding plasma concentration determination, PK/PD parameter calculation and modeling & simulations.
- For instance, in section 2.5-6, large part of modeling and simulation method is missing. Which software was used? What's the final popPK model? How were they developed? What's the final equaiton? How were they validated (any gof graphs etc?) This is crucial for the authencity of the simulations results, so the authors should extend fully on the
-In section 2.4, HPLC-UV model was not mentioned. Also, this section lacks city, country information of the items used.
3. Results
- Results on model validation is not presented in the section.
- The results are presented in a way that could cause misunderstanding. For instance, the legend of Figure 3 explains that the figure is results of 3 g infused over 24h - which should be the result of simulation.? Authors should clarify the real data and the simulated data results.
5. Tables and figures should be mentioned correctly in the manuscript.
- In section 3.2, Figure 1 was not about the estimated PK parameters of MEPM.
- In section 3.3, "The results of 40% fT>MIC,....are shown in Table 3 and Figure 2 and 3. ... (Table 3)", the data in Table 3 is not clearly explained.
Minor
1. English proofreading is recommended to increase the understanding and coherence of the manuscript to the readers.
- Subject is missing in several sentences of manuscript which hinders readers' understanding ( especially section 3.1-2).
2. Authors should make sure that the abbreviations are explained in their first apprearance of the manuscript. In the manuscript, several abbreviations including CVVH, TAM, ..etc are explained only in the table footer, hindering the understanding of manuscript. Authors should make sure that
3. Authors should check if they cited appropriate references. For instance, in introduction it seems that reference 14 was cited inappropriately. Authors should recheck the reference appripriateness throughout the manuscript. Comments on the Quality of English LanguageEnglish proofreading is recommended to increase the understanding and coherence of the manuscript to the readers. Some, but not all, of the comments regarding grammar are the following:
- Subject is missing in several sentences of manuscript which hinders readers' understanding ( especially section 3.1-2).
- "Previous... .regimens. (30)" punctuation missing. run-on sentences.Author Response
Please see the attachment.

Reviewer 2 Report
Comments and Suggestions for Authors
Aggressive PK/PD in Patients Receiving Continuous Renal Replacement Therapy for Septic AKI.” is well written and presents a very impressive and important investigation. This investigation suggests that an MEPM regimen of 1 g infused every 8 h over 3 h for patients receiving CRRT for septic patients achieve a MIC of 0.5 mg/L, and this regimen's risk of toxicity is very low. However, before publishing the manuscript, the author needs to respond to critical queries regarding the research article.
1. Why did the author use the term “Aggressive PK/PD” instead of PK-PD? Please explain aggressive PK/PD in the introduction of the manuscript.
2. Please explain the term “trough level” in the manuscript and how it differs from the PK systemic level.
3. The sample volume is very low (n=9), as the author mentions as a limitation of this study. However, how can the outcome of these data be relevant for determining the regimen for the mass population?
4. Why did the author use “3-(N-morpholino)-propane sulfonic acid (MOPS)” in the plasma collection tube?
5. The author did not mention the extraction procedure” how the MEPM was extracted from plasma. Please include this information in the manuscript.
6. How does the author normalize any error during HPLC analysis without using any internal standard?
7. What is the concentration range of linearity for sample analysis?
8. In Table 1, the number of female subjects is only two. Can this outcome be applied to the entire population? Why did the author not exclude this result?
9. In Table 2, why the p-value of the PK parameter was more than 0.5? How does it significantly correlate to the outcome?
10. In the conclusion section, the author should explain the correlation between eGFR and meropenem clearance, as shown in Figures A and B.
11. In the limitation section, what does the phrase “Second, owing to limitations in insurance reimbursement in Japan, the filtration flow rate for CRRT is not based on body weight” mean? The author needs to improve the English language.
12. In the conclusion section, “achieved up to an MIC of 0.5 mg/L with a MEPM regimen of 1 g infused every 8 h over 3 h for patients receiving CRRT for septic AKI. In addition, this regimen's risk of reaching the toxicity range is low” and is suitable for application. However, the sample size is very low.
13. The manuscript requires an overall improvement in grammar.
Comments on the Quality of English Language13. The manuscript requires an overall improvement in grammar.
Reviewer 3 Report
Comments and Suggestions for Authors
Authors of the manuscript entitled “Association between Extended Meropenem Regimen and Achievement of Aggressive PK/PD in Patients Receiving Continuous Renal Replacement Therapy for Septic AKI ” estimated the model parameters of the Niibe et al.’s popPK model and simulated meropenem concentration-time profiles using a virtual population.
I have several comments below (one major and the other minor ones).
1) The discussion was mainly based on the model simulation. However, the author did not present a measure of model performance. For example, goodness-of-fit plots, absolute error (e.g., mean squared error, and root mean square error), relative error (e.g., fold error, average fold error, absolute average fold error, and root mean squared log error), or fraction of predictions in a two-fold range would be a measure for the purpose. Please consider adding a section to present a step for the model evaluation for this dataset.
2) Species name should be italicized according to the publisher’s guidance (https://www.mdpi.com/authors/layout#_bookmark15). “Pseudomonas aeruginosa” in the introduction section (page 1) was not italicized. Please consider revising.
3) The ‘value’ is defined as “the numerical amount denoted by an algebraic term; a magnitude, quantity, or number” in the Oxford Dictionary. I think the authors were willing to write levels or concentrations rather than ‘value’ in section 3.2 (e.g., steady-state trough concentration and maximum concentration) and Table 2. Please consider revising it.
4) ‘Correlation’ is a statistical measure between two variables. Since the authors mentioned that there was no ‘statistical significance’ between eGFR and CL_total (line 2, page 5), the description might need to be revised. Fig 1a might show a tendency between both parameters instead of ‘correlation’, which is statistical terminology.
5) I think the abbreviation for total volume of distribution was V_total in the main text (page 5), but it was V_body in Table 2. Defining multiple abbreviations for one terminology is confusing. Please consider selecting one of the abbreviations.
6) Please define SOFA and APACHE in the main text (e.g., Sequential Organ Failure Assessment, Acute Physiologic Assessment and Chronic Health Evaluation II, respectively), even though it is defined in the caption of a table.
7) Please define the abbreviation for ‘% fT’ (e.g., the percentage of time that the free concentration).
8) Please define CVVH in the main text, even though it is defined in the caption of a table.
9) Please define KDIGO in the main text.
Comments on the Quality of English LanguageNothing
Round 2
Reviewer 1 Report
Comments and Suggestions for Authors
The authors have sufficiently addressed the concerns raised by the reviewer previously.
Reviewer 3 Report
Comments and Suggestions for Authors
All I commented on have been revised, and I have no additional comments.